METHODS AND PROTOCOLS

# Two-Target Quantitative PCR To Predict Library Composition for Shallow Shotgun Sequencing

Matthew Y. Cho,[a,b,h] Marc Oliva,[c,d] Anna Spreafico,[c] Bo Chen,[e] Xu Wei,[e] Yoojin Choi,[a,b,h] Rupert Kaul,[a,b,h] Lillian L. Siu,[c]
ⓘ Bryan Coburn,[a,b,h] ⓘ Pierre H. H. Schneeberger[a,b,f,g,h]

[a]Department of Medicine, University of Toronto, Toronto, Canada
[b]Department of Medicine, Division of Infectious Diseases, University Health Network, Toronto, Canada
[c]Division of Medical Oncology and Hematology, Princess Margaret Cancer Centre, University of Toronto, Toronto, Canada
[d]Department of Medical Oncology, Catalan Institute of Oncology (Hospital Duran i Reynals), IDIBELL, Barcelona, Spain
[e]Department of Biostatistics, Princess Margaret Cancer Centre, University of Toronto, Toronto, Canada
[f]Department of Medical Parasitology and Infection Biology, Swiss Tropical and Public Health Institute, Basel, Switzerland
[g]University of Basel, Basel, Switzerland
[h]Department of Laboratory Medicine & Pathobiology, University of Toronto, Toronto, Canada

Bryan Coburn and Pierre H. H. Schneeberger are co-senior authors.

**ABSTRACT** When determining human microbiota composition, shotgun sequencing is a powerful tool that can generate high-resolution taxonomic and functional information at once. However, the technique is limited by missing information about host-to-microbe ratios observed in different body compartments. This limitation makes it difficult to plan shotgun sequencing assays, especially in the context of high sample multiplexing and limited sequencing output and is of particular importance for studies employing the recently described shallow shotgun sequencing technique. In this study, we evaluated the use of a quantitative PCR (qPCR)-based assay to predict host-to-microbe ratio prior to sequencing. Combining a two-target assay involving the bacterial 16S rRNA gene and the human beta-actin gene, we derived a model to predict human-to-microbe ratios from two sample types, including stool samples and oropharyngeal swabs. We then validated it on two independently collected sample types, including rectal swabs and vaginal secretion samples. This assay enabled accurate prediction in the validation set in a range of sample compositions between 4% and 98% nonhuman reads and observed proportions varied between $-18.8\%$ and $+19.2\%$ from the expected values. We hope that this easy-to-use assay will help researchers to plan their shotgun sequencing experiments in a more efficient way.

**IMPORTANCE** When determining human microbiota composition, shotgun sequencing is a powerful tool that can generate large amounts of data. However, in sample compositions with low or variable microbial density, shallowing sequencing can negatively affect microbial community metrics. Here, we show that variable sequencing depth decreases measured alpha diversity at differing rates based on community composition. We then derived a model that can determine sample composition prior to sequencing using quantitative PCR (qPCR) data and validated the model using a separate sample set. We have included a tool that uses this model to be available for researchers to use when gauging shallow sequencing viability of samples.

**KEYWORDS** shotgun sequencing, shallow shotgun, microbiome, sample composition, host DNA proportion, metagenomics

Address correspondence to Bryan Coburn, bryan.coburn@utoronto.ca, or Pierre H. H. Schneeberger, pierre.schneeberger@swisstph.ch.

**S**hotgun sequencing allows interrogation of the metagenomic composition of ecological niches and has been increasingly utilized to characterize human-associated microbial communities. Shallow shotgun sequencing—sequencing to a per-sample read

depth of $10^5$ to $10^6$ reads—provides taxonomic resolution greater than that of 16S amplicon sequencing and functional characterization of metagenomes, while being less expensive than whole-genome sequencing or deep sequencing (typically $10^7$ to $10^9$ reads/sample) (1). However, there is a trade-off between cost and adequacy, which is especially problematic for samples of variable ratios of host to microbial DNA, where microbial reads may be displaced by human reads in a mixed sample (2). While this is generally not a concern for samples with high bacterial load, such as stool samples, samples with low or variable amounts of microbial DNA relative to that of human DNA are common in other regions of the body, such as the lung, nasopharynx, stomach, and duodenum (2–5). Bacterial density can range depending on sample site from $10^3$ to $10^{11}$ CFU/ml, and ranges within a sample site can vary by up to 4 orders of magnitude (5, 6). Microbial taxonomic and functional analyses of metagenomic data require sufficient reads to draw robust conclusions. The ability to predict the proportion of microbial reads prior to sequencing would allow researchers to customize sequencing strategies for desired analyses while optimizing the cost and time spent on metagenomic sequencing.

In this study, we used quantitative PCR to predict the ratio of human to microbial reads obtained from sequencing using three targets: the 16S rRNA gene, 18S rRNA gene, and human beta-actin (ACTB) to quantitate DNA of bacterial, fungal, and human origin, respectively (7–9). We compared the ratios of bacterial to human DNA determined via quantitative PCR (qPCR) to the percent microbial/human DNA determined via shallow shotgun sequencing in samples with variable bacterial DNA. We derived a prediction model from oropharyngeal swabs and stool samples, and evaluated it in a set of independently collected samples, including rectal swabs and vaginal secretion samples. Finally, we generated an easy-to-use tool based on qPCR data to predict sample composition and sequencing depth required given a desired analytical outcome.

(This article was submitted to an online preprint archive [10].)

## RESULTS AND DISCUSSION

To assess the impact of shallow sequencing depth on different bacterial DNA proportions, we rarefied shotgun sequencing data from 4 sample types—stool, oropharyngeal, rectal, and vaginal—to depths of 1,000 to 1 million reads/sample. We then determined the alpha diversity of each rarefaction using three metrics: richness, Shannon index, and Berger-Parker index. Alpha diversity decreased in a sample type-specific manner as sequencing depth decreased (Fig. 1). Notably, while vaginal samples had the lowest alpha diversity in all three metrics of the four sample types, alpha diversity decreased at the lowest rate as sequencing depth decreased (Fig. 1). Conversely, while rectal swab samples had similar Shannon index and Berger-Parker index values at $10^6$ microbial reads to oropharyngeal and stool samples, alpha diversity in rectal samples diminished at a greater rate as sequencing depth decreased (Fig. 1B and C). Since this effect is sample-type specific, it is critical to predict sample composition *a priori* to ensure sufficient reads for the desired analysis for the given sample type. In addition, we observed that both microbial DNA content and total DNA density can greatly vary even within the same sample type, and as such, sample composition should be determined on a sample-by-sample basis (see also Fig. S1 in the supplemental material).

qPCR is a widespread and robust technique available in many molecular biology laboratories. Its availability as well as inexpensive associated costs, especially compared to that for experiments involving high-throughput sequencing techniques, makes it an ideal candidate to use to predict sample composition prior to sequencing. In this study, we assessed the potential of qPCR to predict sample-specific ratios of human to microbe DNA using different amplification targets. Using a multivariate approach, 5 models were derived mapping the rRNA gene, 18S rRNA gene, and human beta-actin (ACTB) qPCR-derived cycle thresholds (Ct) to observe the percentage of microbial reads for a sample set consisting of oropharyngeal swabs and stool samples. Microbial reads were defined as any read which did not align/match with a human genome reference. The following models were tested: (A) a linear fit to the percent microbial reads using

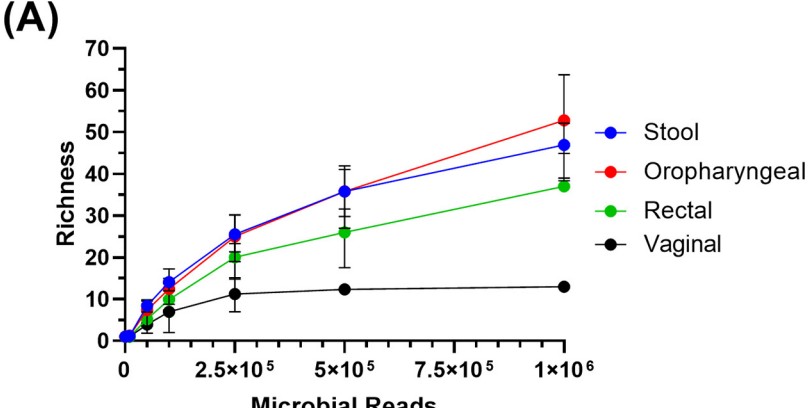

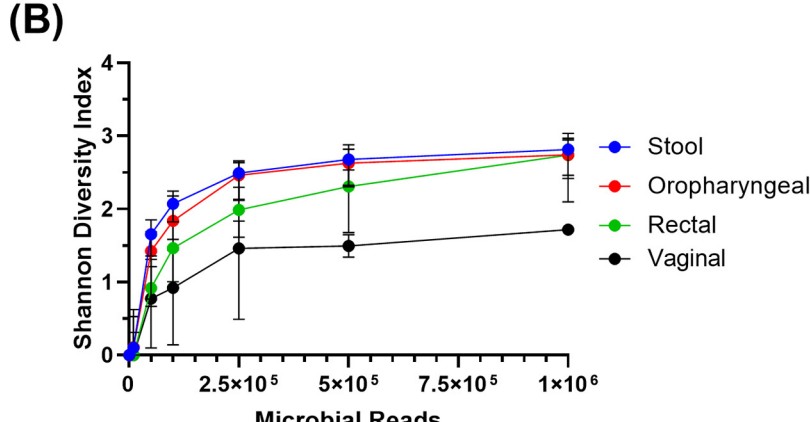

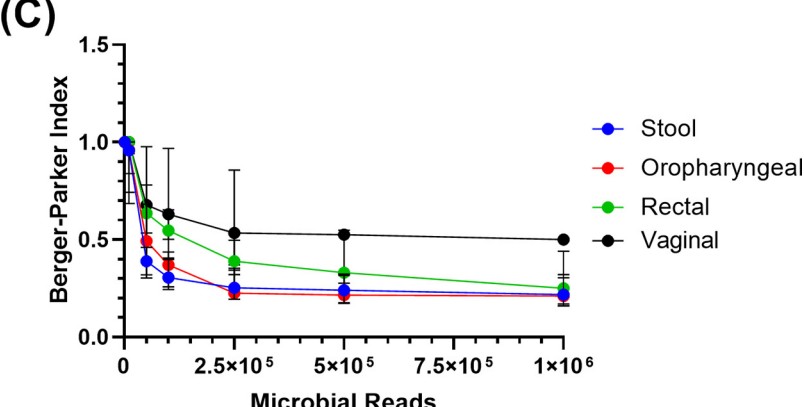

**FIG 1** Alpha diversity indices are shown across a range of simulated sequencing depths from $10^3$ to $10^6$ reads per sample. Each sample was subsampled 10 times for a range of sequencing depths. Each resulting rarefaction was profiled using MetaPhlAn 2.0. Richness, Shannon index, and Berger-Parker indexes were calculated for each rarefaction. The mean value of each index was calculated per sample per depth. Displayed are the median values and interquartile ranges of these means by sample origin. (A) Sample-specific rarefaction curves of species richness. (B) Shannon index calculated across a range of rarefactions, by sample type. (C) Sample dominance, measured with the Berger-Parker index, across a range of sequencing depths, stratified by sample type.

16S rRNA gene and *ACTB* Ct values, (B) a linear fit to the percent microbial reads using 16S rRNA gene, 18S rRNA gene, and *ACTB* Ct values, (C) a linear fit to logit-transformed percent microbial reads using 16S rRNA gene and *ACTB* Ct values, (D) a linear fit to logit-transformed percent microbial reads using 16S rRNA gene, 18S rRNA gene, and *ACTB* Ct values, and (E) a nonlinear regression model based on the logistic growth equation using 16S rRNA gene and *ACTB* Ct values (Fig. 2A and S2). We compared the

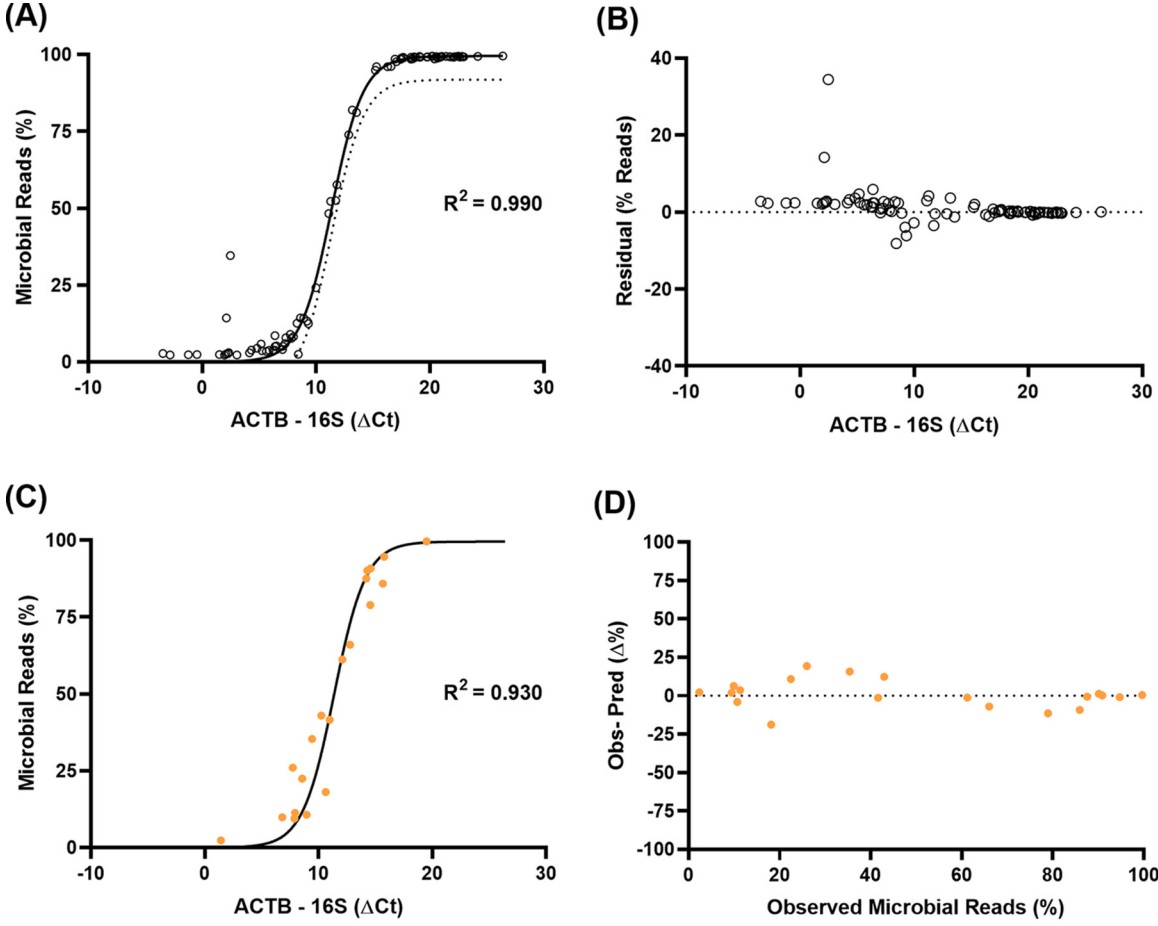

**FIG 2** Statistical models to predict sample composition using qPCR prior to high-throughput sequencing. (A) Sigmoidal model derived from oropharyngeal swabs and stool samples depicting the relationship between the difference of human (*ACTB*) and bacterial (16S) qPCR values (Ct) with the percentage of microbial reads ($R^2 = 0.990$). Nonlinear regression line (solid) is based on the following logistic growth equation: percent microbial reads = $2.7201549/([99.50267 \times e^{-0.7218 \times \{ACTB - 16S\}}] + 0.02733)$. One-tailed 95% prediction interval is depicted with a dotted line. (B) Model residuals. (C) Fitting of validation sample set on prediction model. The orange dots represent values derived from a validation sample set composed of vaginal secretion and rectal swab samples and correlate well ($R^2 = 0.930$) with the prediction model (solid black line). (D) Difference between expected and observed composition across the range of microbial content.

goodness of fit for each model and observed $R^2$ values of 0.880, 0.880, 0.920, 0.920, and 0.990 for models A to E, respectively (Fig. 2A and S2). Observed residuals had a minimum to maximum range of 67.56, 68.50, 58.93, 59.07, and 42.61 for models A to E, respectively (see Fig. S3A; Table S1A). Based on these findings, model E turned out to be the best fitting model to predict sample composition using qPCR, with the equation percent microbial reads = $2.7201549/([99.50267 \times e^{-0.7218 \times \{ACTB - 16S\}}] + 0.02733)$. In addition, 18S rRNA Ct value was not found to be an informative predictor and was hence removed from the model. In Fig. 2B, we show the goodness of fit and residuals observed with model E across the range of qPCR differences (−8.16% to +34.45%). We observed homogeneous fit and variance, indicating that the model performs well for all observed host-to-microbe DNA ratios. However, we also observed that due to the asymptotes inherent to the sigmoidal model along with limitations of the derivation data set, the model loses accuracy for samples with observed microbial reads less than 4% or greater than 98%. This bias is likely introduced at different steps of the process. For instance, sequencing error and resulting false-negative and -positive hits when mapping reads to the human database are likely to account for this bias. Another potential source of bias could be introduced by the carryover of contaminants between sequencing runs, hence resulting in a composition change which is not picked up by the qPCR conducted *a priori*.

Using the equation derived from model E, we evaluated our approach on two different independently collected sample types, including vaginal secretions and rectal swabs. In Fig. 2C, we show the relation between observed microbial read percentages and the difference in Cts between 16S and *ACTB* qPCR, derived from our validation data set, alongside a curve of expected values derived from model E. We observed the difference between predicted and observed microbial reads percentages to range from $-18.80\%$ to $+19.22\%$ with a mean of $+0.944\%$ (Fig. S3B). In Fig. 2D, we show that this difference is consistent across the range of observed percent microbial reads. Compared to the other models, model E best described the validation data set, with a median difference of 0.25% and a standard deviation of 9.10% (Table S1B; Fig. S2). For comparison, model E described the initial sample set of oropharyngeal/stool samples with a median difference of 0.14% and a standard deviation of 4.35% (Table S1A). Since the model performed similarly between the two data sets, we concluded that the model was able to describe a relationship between 16S and $\beta$-actin qPCR results and the composition of shotgun sequencing metagenomic data in a sample-type-independent manner for microbial proportions comprising between 4% and 98%. We then developed a tool based on our model and the rarefaction curves on different samples types which predicts percent of microbial reads based on qPCR data and suggests a target number of reads based on sample type and desired analysis (see Table S4).

However, models C and D shared very similar range and standard deviation values with model E, while arguably being more accessible, being multivariate linear models of logit-transformed values (Table S1B). These models were shifted such that the mean residual value was zero (see Fig. S4A). The distribution of the absolute value of the residuals suggests that shifting models C and D improved precision but were still not as precise as model E (Fig. S4B). Therefore, models C and D may be options for researchers that consider the trade-off of precision for accessibility worthwhile.

The limitations of our study are as follows. The samples used in our study were low in fungal content. Therefore, our model may not accurately predict microbial content in sample sets where the majority of samples are rich in fungal content. In addition, *in silico* analysis shows that our bacterial 16S primer set provides broad coverage of phyla most common in human samples: *Proteobacteria*, *Actinobacteria*, *Firmicutes*, and *Bacteroidetes* (11). However, there is no coverage of the phyla *Spirochaetes* and *Chlamydiae* (11). As such, we may be underreporting bacterial density in sites where these phyla are more common (12).

Moreover, as our results are based on protocols using specific reagents and technologies for both sequencing and qPCR, our tool may not accurately predict sequencing results when protocols, reagents, and/or technologies differ. However, given that we have established a robust link among 16S qPCR, $\beta$-actin qPCR, and sample content by sequencing, our approach can be easily adapted to fit different experimental settings.

Our approach does not replace the need to prespecify sequencing requirements for a given application. The use of qPCR is intended to complement small exploratory experiments that establish required depths when scaling up metagenomic sequencing to larger projects, and in this way, we feel it adds significant potential use value.

**Conclusion.** We have shown that a shallow shotgun sequencing depth can reduce measured alpha diversity in all measured sample types, with more diverse communities being more strongly negatively affected. We found that qPCR can function as a predictive tool for sample composition that was strongly correlated with shotgun sequencing data. We were able to create a model that can describe and predict variable sample types. We hope that our tool and methodology may help fellow researchers screen for sequenceable samples or allow for better optimization of sequencing.

## MATERIALS AND METHODS

**Sample acquisition and preparation.** Oropharyngeal swabs and stool samples were collected from a cohort of patients with human papillomavirus-positive (HPV$^+$) locoregionally advanced oropharyngeal squamous cell carcinoma (LA-OPSCC) treated with chemoradiotherapy (CRT) (13). Oropharyngeal swabs over the tumor site and stool samples were collected up to 3 weeks prior to the start of radiotherapy and again at completion of CRT (up to 3 weeks following last day of radiotherapy) (13). Oropharyngeal swabs over the tumor site and stool samples were collected using the ZymoBIOMICS DNA/RNA mini prep kits (Zymo Research, Irvine, CA). DNA was extracted using ZymoBIOMICS DNA micro kit (13).

Vaginal secretions were collected from a cohort of patients 10 to 18 days after the last day of bleeding from their previous menstrual period (11). Instead SoftCups (Evofem, San Diego, CA) were self-inserted to collect undiluted cervicovaginal secretions for 1 min (14). The SoftCup was placed into a 50-ml conical tube and transported on ice to the lab within 2 h (14). DNA was extracted using the DNeasy PowerSoil kit (Qiagen) (14).

Rectal swabs (FLOQSwab; Copan) were collected from a cohort of HIV-positive, antiretroviral therapy (ART)-treated men who have sex with men in Toronto, Canada (15). Rectal swabs were inserted and rotated 360° inside the anal canal (15). All rectal swabs were stored at −80°C in 300 $\mu$l of ultrapure-grade phosphate-buffered saline (VWR, Radnor, USA) prior to DNA extraction (15). DNA was extracted using the DNeasy PowerSoil kit (Qiagen) per the manufacturer's instructions with one exception: 30 $\mu$l of solution C1 was used to treat both the supernatant and the swab for the first step before removing the swab and adding another 30 $\mu$l of C1 to complete the process (15).

All studies were approved by the institutional research ethics board. All patients provided written, signed, informed consent to participate. In total, 46 oropharyngeal swabs, 46 stool samples, 7 vaginal samples, and 13 rectal swabs were included in this study. Two oropharyngeal swabs and 1 stool sample were not taken into consideration in model derivation due to insufficient amounts of sample for qPCR. Oropharyngeal and stool samples were grouped into a sample set used to derive models, while vaginal and rectal samples comprised the sample set used to validate.

**qPCR.** Samples were probed separately for the 16S rRNA gene, the 18S rRNA gene, and the human $\beta$-actin gene. All reactions were conducted in duplicates, and RNase-free water was used as a negative control. Each well contained 2 $\mu$l of sample DNA, 5 $\mu$l of TaqMan universal PCR mix (Applied Biosystems, Foster City, CA), 0.3 $\mu$M forward primer, 0.3 $\mu$M reverse primer, and 0.2 $\mu$M primer probe with required distilled water for a total volume of 10 $\mu$l per well. PCR was performed on a QuantStudio 6 Flex (Thermo Fisher Scientific, Waltham, MA) platform. Cycling was conducted as follows: 10 min at 95°C followed by 45 cycles of 95°C for 15 s and 60°C for 1 min.

To determine bacterial DNA content, we used primers and probes specific for the bacterial 16S rRNA gene designed by Nadkarni et al. (7) (see Table S1 in the supplemental material). Their primer set was able to detect 34 different species from various genera while avoiding cross-detection of DNA from kingdoms *Eucarya* and *Archaea* (7).

To determine fungal DNA content, we used the FungiQuant primer set designed by Liu et al. (8), due to its high coverage and specificity for fungal 18S rRNA gene sequences (Table S1). Liu et al. found that *in silico*, the primer had a perfect match for 91.4% of genera of 18 subphyla and could be used to accurately measure fungal 18S content up to ratios of 25:679,464 fungal-to-human 18S rRNA gene copy number (8).

To determine human DNA content, we used a set of $\beta$-actin gene-specific primers and probes designed by Hasan et al. (9) (Table S1).

**Library preparation and sequencing.** Libraries were prepared using Nextera Flex (Illumina, San Diego, CA) kits with the Nextera XT indices (Illumina). Barcoded sample libraries were pooled to a concentration of 17.6 ng/$\mu$l measured with a high-sensitivity DNA assay on a Qubit (Thermo Fisher Scientific, Waltham, MA) platform. A mid-output reagent kit (Illumina) was used to sequence on the MiniSeq, while an SP reagent kit (Illumina) was used on the NovaSeq platform, both in 2 by 150-bp mode.

**Read filtering and taxonomic profiling.** We filtered human reads from nonhuman reads using KneadData based on a human genome index for Bowtie 2 (16, 17). We considered sequence reads that did not match the database as microbial reads in our analyses. Taxonomic annotation was conducted using MetaPhlAn 2.0 and the ChocoPhlAn database (18). Rarefactions were performed using seqtk-1.3 to subsample the microbial reads of individual samples (19). Each sample was subsampled at depths of 1 million, 500,000, 250,000, 100,000, 50,000, 10,000, and 1,000 reads 10 times using 10 distinct seeds. Subsample compositions were identified using MetaPhlAn 2.0, and taxonomic profiles were generated (18). Richness, Shannon index, and Berger-Parker dominance index were calculated using Past 4 for each rarefaction (20). The mean value of each index was calculated per sample for the range of sequencing depths. The median value of each index per sample site was then calculated for the range of sequencing depths (Fig. 1).

**Model derivation.** We used XLSTAT version 2019.4.2 (Addinsoft Inc., New York, NY) to derive multivariate linear regressions using either 16S and *ACTB* qPCR cycle and microbial read percentages (models A and C) or 16S, 18S, and *ACTB* qPCR cycle thresholds and microbial read percentages (models B and D). Multivariate linear regressions (models C and D) were also performed following a logit transformation of microbial read percentages. Finally, for model E, we derived the nonlinear regression model using the logistic growth equation in GraphPad Prism version 8.3.0 for Windows (GraphPad Software, San Diego, CA).

**Data availability.** Raw sequencing data have been uploaded to the Short Read Archive (SRA) under accession number PRJNA718445 as a series of fastq files. Six patients from the stool/oropharyngeal cohort declined to have their data made publicly available. Exact status of sequencing data availability and raw qPCR cycle thresholds can be found in Table S3. To reproduce our results, raw sequencing data must be filtered for human reads using KneadData or similar software to determine percent microbial reads per sample by comparing the number of reads pre- and postfiltering.

## SUPPLEMENTAL MATERIAL

Supplemental material is available online only.

**FIG S1**, TIF file, 0.2 MB.

**FIG S2**, TIF file, 0.4 MB.

**FIG S3**, TIF file, 0.3 MB.

**FIG S4**, TIF file, 0.3 MB.

**TABLE S1**, PDF file, 0.1 MB.
**TABLE S2**, PDF file, 0.1 MB.
**TABLE S3**, PDF file, 0.1 MB.
**TABLE S4**, XLSX file, 0.1 MB.

## ACKNOWLEDGMENTS

This work was supported by grants from Ontario Genomics, Physician Services Inc. Foundation, The Canadian Cancer Society, The Tomcyzk AI and Microbiome Working Group, and The Princess Margaret Cancer Foundation. P.H.H.S. was supported with a fellowship from the Tomcyzk AI and Microbiome Working Group and The Princess Margaret Cancer Foundation.

M. Oliva is a compensated consultant for Bristol-Myers Squibb Canada and Mirati Therapeutics and has received grant/research support (clinical trials) from Mirati Therapeutics and NuBiyota. L.L. Siu is a compensated consultant for Merck, Pfizer, Celgene, AstraZeneca/Medimmune, MorphoSys, Roche, GeneSeeq, Loxo, Oncorus, Symphogen, Seattle Genetics, GSK, Voronoi, and Treadwell Therapeutics and has received grant/research support (clinical trials for institution) from Novartis, Bristol-Myers Squibb, Pfizer, Boehringer-Ingelheim, GlaxoSmithKline, Roche/Genentech, Karyopharm, AstraZeneca/Medimmune, Merck, Celgene, Astellas, Bayer, AbbVie, Amgen, Symphogen, Intensity Therapeutics, Mirati, Shattuck Labs, and Avid. L.L. Siu is a stockholder of Agios. A. Spreafico is a compensated consultant for Merck, Bristol-Myers Squibb, Novartis, and Oncorus and has received grant/research support (clinical trials) from Novartis, Bristol-Myers Squibb, Symphogen AstraZeneca/Medimmune, Merck, Bayer, Surface Oncology, Northern Biologics, Janssen Oncology/Johnson & Johnson, Roche, and Array Biopharma. All other authors declare that they have no competing interests.

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
