## [Reviewer comments · mSystems]

Two-target quantitative PCR to predict library composition for shallow shotgun sequencing

Matthew Cho, Marc Oliva, Anna Spreafico, Bo Chen, Xu Wei, Yoojin Choi, Rupert Kaul, Lillian Siu, Bryan Coburn, and Pierre Schneeberger

Corresponding Author(s): Bryan Coburn, University Health Network

Review Timeline:

Submission Date:	May 4, 2021
Editorial Decision:	June 6, 2021
Revision Received:	June 16, 2021
Accepted:	June 17, 2021

Editor: Anthony Fodor

Reviewer(s): The reviewers have opted to remain anonymous.

Transaction Report:

DOI: <https://doi.org/10.1128/mSystems.00552-21>

June 6, 2021

Dr. Bryan Coburn
University Health Network
Toronto, Ontario M5G 1L7
Canada

Re: mSystems00552-21 (Two-target quantitative PCR to predict library composition for shallow shotgun sequencing)

Dear Dr. Bryan Coburn:

Thank you for submitting your manuscript to mSystems. Your manuscript was reviewed by one of the original reviewers as well as two members of the editorial board (including myself). I am pleased to inform you that, in principle, we expect to accept it for publication in mSystems. However, acceptance will not be final until you have adequately addressed the following comments:

Reviewer #2 had a suggestion for the discussion. Please incorporate the text from your response into the discussion as noted below.

A large part of the contribution of the manuscript is the dataset. Please add an explicit "Data Availability" paragraph at the end of the Materials and Methods section that includes a data description, name of the repository, and the accession numbers (which I believe are currently in the response to reviewers but not in the manuscript). Please ensure that the data availability section includes access to the qPCR data as well as the sequence data and has explicit instructions on how to link the qPCR identifiers to sequence identifiers (as much as is possible given the limitation that not all of your subjects allowed sequence release). You could consider adding supplementary directions on how to download and link the qPCR and sequence data if this is more detailed than can be addressed in the "Data Availability" paragraph. These instructions would presumably be more explicit than the UNIX code fragment currently present in the supplementary materials in allowing users to reproduce the figures in the paper.

Preparing Revision Guidelines

To submit your modified manuscript, log onto the eJP submission site at <https://msystems.msubmit.net/cgi-bin/main.plex>. Go to Author Tasks and click the appropriate manuscript title to begin the revision process. The information that you entered when you first submitted the paper will be displayed. Please update the submission to address these last concerns. Please let me know if you have questions.

For complete guidelines on revision requirements, please see the Instructions to Authors at <https://msystems.asm.org/sites/default/files/additional-assets/mSys-ITA.pdf>. **Submissions of a paper that does not conform to mSystems guidelines will delay acceptance of your manuscript.**

Corresponding authors may join or renew ASM membership to obtain discounts on publication fees. Need to upgrade your membership level? Please contact Customer Service at

Service@asmusa.org.

Sincerely,

Anthony Fodor

Editor, mSystems

Journals Department
Reviewer comments:

Reviewer #2 (Comments for the Author):

I am satisfied with the authors' current responses to my comments. Still, I would encourage the authors to also consider adding to their discussion the comment they provided in their response to me: "Our approach doesn't replace the need to prespecify sequencing requirements for a given application. The use of qPCR is intended to complement small exploratory experiments that establish required depths when scaling up metagenomic sequencing to larger projects, and in this way." Such insight I think will be valuable to future users.

Dear Dr. Bryan Coburn:

Thank you for submitting your manuscript to mSystems. Your manuscript was reviewed by one of the original reviewers as well as two members of the editorial board (including myself). I am pleased to inform you that, in principle, we expect to accept it for publication in mSystems. However, acceptance will not be final until you have adequately addressed the following comments:

Reviewer #2 had a suggestion for the discussion. Please incorporate the text from your response into the discussion as noted below.

A large part of the contribution of the manuscript is the dataset. Please add an explicit "Data Availability" paragraph at the end of the Materials and Methods section that includes a data description, name of the repository, and the accession numbers (which I believe are currently in the response to reviewers but not in the manuscript). Please ensure that the data availability section includes access to the qPCR data as well as the sequence data and has explicit instructions on how to link the qPCR identifiers to sequence identifiers (as much as is possible given the limitation that not all of your subjects allowed sequence release). You could consider adding supplementary directions on how to download and link the qPCR and sequence data if this is more detailed than can be addressed in the "Data Availability" paragraph. These instructions would presumably be more explicit than the UNIX code fragment currently present in the supplementary materials in allowing users to reproduce the figures in the paper.

We have added an explicit “Data Availability” paragraph at L284-292, which includes the file type of the sequencing data, the name of the repository, the accession number, and a hyperlink to the sequencing data. We have also made the raw qPCR Ct values accessible in the paper in Supplementary table 3, L345-347.

Preparing Revision Guidelines

To submit your modified manuscript, log onto the eJP submission site at <https://msystems.msubmit.net/cgi-bin/main.plex>. Go to Author Tasks and click the appropriate manuscript title to begin the revision process. The information that you entered when you first submitted the paper will be displayed. Please update the submission to address these last concerns. Please let me know if you have questions.

For complete guidelines on revision requirements, please see the Instructions to Authors at <https://msystems.asm.org/sites/default/files/additional-assets/mSys-ITA.pdf>. Submissions of a paper that does not conform to mSystems guidelines will delay acceptance of your manuscript.

Sincerely,

Anthony Fodor

Editor, mSystems

Journals Department
Reviewer comments:

Reviewer #2 (Comments for the Author):

I am satisfied with the authors' current responses to my comments. Still, I would encourage the authors to also consider adding to their discussion the comment they provided in their response to me: "Our approach doesn't replace the need to prespecify sequencing requirements for a given application. The use of qPCR is intended to complement small exploratory experiments that establish required depths when scaling up metagenomic sequencing to larger projects, and in this way." Such insight I think will be valuable to future users.

The above paragraph has been added to the manuscript within the discussion at L193-196.

June 17, 2021

Dr. Bryan Coburn
University Health Network
Toronto, Ontario M5G 1L7
Canada

Re: mSystems00552-21R1 (Two-target quantitative PCR to predict library composition for shallow shotgun sequencing)

Dear Dr. Bryan Coburn:

Your manuscript has been accepted, and I am forwarding it to the ASM Journals Department for publication. For your reference, ASM Journals' address is given below. Before it can be scheduled for publication, your manuscript will be checked by the mSystems senior production editor, Ellie Ghatineh, to make sure that all elements meet the technical requirements for publication. She will contact you if anything needs to be revised before copyediting and production can begin. Otherwise, you will be notified when your proofs are ready to be viewed.

We recognize that the video files can become quite large, and so to avoid quality loss ASM suggests sending the video file via <https://www.wetransfer.com/>. When you have a final version of the video and the still ready to share, please send it to Ellie Ghatineh at eghatineh@asmusa.org.

Sincerely,

Anthony Fodor
Editor, mSystems

Journals Department
Supplementary table 2: Accept
Supplemental Material: Accept
Supplementary table 1: Accept
Supplementary figure 2: Accept
Supplementary figure 1: Accept
Supplementary Tool: Accept
Supplementary figure 3: Accept
Supplementary figure 4: Accept